# FACT: A FIRST-PRINCIPLES ALTERNATIVE TO THE NEURAL FEATURE ANSATZ FOR HOW NETWORKS LEARN REPRESENTATIONS

**Enric Boix-Adsera**[*]
UPenn

**Neil Mallinar**[*]
UCSD

**James B. Simon**
Imbue, UC Berkeley

**Mikhail Belkin**
UCSD

## ABSTRACT

It is a central challenge in deep learning to understand how neural networks learn representations. A leading approach is the Neural Feature Ansatz (NFA) (Radhakrishnan et al., 2024), a conjectured mechanism for how feature learning occurs. Although the NFA is empirically validated, it is an educated guess and lacks a theoretical basis, and thus it is unclear when it might fail, and how to improve it. In this paper, we take a first-principles approach to understanding why this observation holds, and when it does not. We use first-order optimality conditions to derive the Features at Convergence Theorem (FACT), an alternative to the NFA that (a) obtains greater agreement with learned features at convergence, (b) explains why the NFA holds in most settings, and (c) captures essential feature learning phenomena in neural networks such as grokking behavior in modular arithmetic and phase transitions in learning sparse parities, similarly to the NFA. Thus, our results unify theoretical first-order optimality analyses of neural networks with the empirically-driven NFA literature, and provide a principled alternative that provably and empirically holds at convergence.

## 1 INTRODUCTION

A central aim of deep learning theory is to understand how neural networks learn representations. An empirically-driven conjecture that has recently emerged as to the mechanism driving feature learning in neural networks is the Neural Feature Ansatz (NFA) (Radhakrishnan et al., 2024), which states that, after training, a weight layer $W$ in a neural network $f(x)$ satisfies the proportionality relation

$$W^\top W \propto \hat{\mathbb{E}}[(\nabla_x f(x))(\nabla_x f(x))] \,.$$

Here the right-hand-side is an empirical expectation over the training data, and the gradient is computed with respect to the input to the network (or with respect to the hidden activations if $W$ is not the first layer) – we review more details on the NFA conjecture later, in Section 2.

This conjecture has been validated in practice on a range of architectures, including fully-connected networks, convolutional networks, and transformers (Radhakrishnan et al., 2024). Furthermore, a growing literature has shown that this NFA conjecture captures and explains several intriguing phenomena of neural network training, including grokking of modular arithmetic (Mallinar et al., 2025), learning of hierarchical staircase functions (Zhu et al., 2025), and catapult spikes during training (Zhu et al., 2023). Additionally, when used to power an adaptive kernel learning algorithm, it achieves state-of-the-art performance for monitoring models (Beaglehole et al., 2025), for learning tabular datasets (Radhakrishnan et al., 2024), and for low-rank matrix learning (Radhakrishnan et al., 2025).

Despite its success, the NFA conjecture lacks first-principles backing for why it should necessarily hold during training. Because this conjecture was derived in an empirical fashion, it is unclear why it ought to hold, whether and under which conditions it may fail, and how to improve it. This motivates the main question studied by this paper:

*Is there an alternative to the empirically-observed Neural Feature Ansatz conjecture, which can be derived from first principles?*

We answer this question in the affirmative. Our main contribution is to demonstrate a connection between the empirically-observed NFA conjecture and the literature studying first-order optimality conditions that must provably hold if the training process converges.

First-order optimality conditions have previously been shown to imply several phenomena in neural network training; see the related work in Section 1.1 for references. Thus, **our results unify two prominent approaches to studying feature learning (the NFA and first-order optimality)**. In more detail, our contributions are:

(1) **We derive a simple alternative to the NFA conjecture based on first-order optimality conditions**. We call this the *FACT (Features at Convergence Theorem)*. This is a self-consistency formula that neural networks trained with weight decay must satisfy at convergence; see Section 3.

(2) **We empirically demonstrate that our first-principles alternative captures neural network feature learning phenomena in many of the same ways that the NFA conjecture does.** We show that when FACT (instead of NFA) is used to power an adaptive kernel learning algorithm (Radhakrishnan et al., 2024), it also reproduces intriguing feature learning behaviors observed in neural networks such as training phase transitions when learning sparse parities (Barak et al., 2022; Abbe et al., 2023), grokking of modular arithmetic (Nanda et al., 2023; Gromov, 2023), and high performance on tabular data matching the state-of-the-art (Radhakrishnan et al., 2024); see Section 4.

(3) **We provide a derivation for why the NFA conjecture usually holds based on first-order optimality.** By algebraically expanding the FACT relation, and analyzing the terms, we demonstrate that it is qualitatively similar to the conjectured NFA relation. We empirically demonstrate that the two relations are proportional in the case of modular arithmetic. This helps put the NFA conjecture on firm theoretical foundation by connecting it to provable first-order optimality conditions, and elucidates the mystery of why it usually holds; see Section 5.

(4) **We construct degenerate training settings in which the NFA conjecture is provably false but where first-order optimality conditions hold true.** We formally prove and experimentally observe that in certain settings the NFA predictions can be nearly uncorrelated to the ground truth, while FACT and any other relations based on first-order-optimality conditions still hold. This indicates that the latter may provide a more accurate relation at convergence; see Section 6 as well the discussion in Section 7.

## 1.1 RELATED LITERATURE

**Implications of first-order-optimality in neural networks** First-order optimality conditions of networks at convergence – along with results on KKT conditions that arise with exponentially-tailed losses at large training times (Soudry et al., 2018; Ji & Telgarsky, 2019; Lyu & Li, 2019; Ji & Telgarsky, 2020) – has been used to show implicit bias of deep architectures towards low rank (Gunasekar et al., 2017; Arora et al., 2019b; Galanti et al., 2022), of diagonal networks towards sparsity (Woodworth et al., 2020), of convolutional networks towards Fourier-sparsity (Gunasekar et al., 2018), and of fully-connected networks towards algebraic structure when learning modular arithmetic (Mohamadi et al., 2023; Morwani et al., 2023). First-order optimality also has implications to linear regression with bagging (Stewart et al., 2023), understanding adversarial examples (Frei et al., 2024), and neural collapse (Han et al., 2021; Kothapalli, 2022; Zangrando et al., 2024).

**Analyses of training dynamics** Another recently prevalent approach to understanding feature learning is to study neural network training dynamics – tracking weight evolution to understand how features emerge (Olsson et al., 2022; Edelman et al., 2024; Nichani et al., 2024; Cabannes et al., 2023; 2024; Arous et al., 2021; Abbe et al., 2022; 2023; Kumar et al., 2023). While insightful, these analyses are technically challenging and are typically limited to synthetic datasets. In this paper, we pursue an alternative approach: we seek conditions on network weights that are satisfied *at the conclusion of training*, to gain insight into how the trained network represents the learned function. By focusing on the network state at convergence, we can circumvent many of the difficulties associated with analyzing training dynamics, and the insights directly apply beyond simplified synthetic settings.

Figure 1: The model only depends on $W$ through multiplication of activations $h(x)$.

**Equivariant NFA**   Another alternative to the NFA, called the "equivariant NFA" (eNFA), was recently proposed in Ziyin et al. (2025) based on an analysis of the dynamics of noisy SGD, which is invariant to linear transformations in the loss function. This is distinct from the FACT and we also compare to it in Section 4.

**Relation to literature on representation identifiability**   The focus of this paper is distinct from the literature on representation identifiability and Independent Component Analysis (ICA), see e.g. (Hyvärinen et al., 2024) and references therein. The identifiability literature is primarily concerned with the *inverse problem*: determining the sufficient conditions (such as non-Gaussianity or the presence of auxiliary variables) under which the true latent generative factors can be uniquely recovered from observed data. In contrast, this paper addresses the structure of representations learned by neural networks.

## 2   TRAINING SETUP AND BACKGROUND

**Training setup**   We consider the standard training setup, with a model $f(\cdot; \theta) : \mathcal{X} \to \mathbb{R}^c$ trained on a sample-wise loss function $\ell : \mathbb{R}^c \times \mathcal{Y} \to \mathbb{R}$ on data points $(x_i, y_i)_{i \in [n]}$ with $L^2$ regularization parameter $\lambda > 0$ (that is, with non-zero weight decay). The training loss is $\mathcal{L}_\lambda(\theta) = \mathcal{L}(\theta) + \frac{\lambda}{2}\|\theta\|_F^2$, where $\mathcal{L}(\theta) = \frac{1}{n}\sum_{i=1}^n \ell(f(x_i; \theta), y_i)$. Here $\mathcal{X}$ and $\mathcal{Y}$ are the input and output domains.

Our FACT applies to any weight matrix parameter $W \in \mathbb{R}^{d' \times d}$ inside a trained model. The only architectural requirement is that the model only depends on $W$ via matrix multiplication of internal activations. See Figure 1. Formally, fixing all parameters but $W$, there are functions $g, h$ such that for all $x$,

$$f(x; \theta) = g(Wh(x), x). \tag{2.1}$$

In this notation,[1] $h$ is the input to the weight matrix, and $Wh$ is the output. Thus, FACT applies to any layer in neural networks that involves matrix multiplications.

For convenience, we introduce the notation to denote the gradient of the loss and the value of the model *with respect to the input of the layer* containing the weight matrix $W$, at the data point $x_i$:

$$\nabla_h \ell_i := \frac{\partial \ell(g(Wh, x); y_i)}{\partial h} \Big|_{h = h(x_i)} \in \mathbb{R}^d \quad \text{and} \quad \nabla_h f_i := \frac{\partial g(Wh, x)}{\partial h} \Big|_{h = h(x_i)} \in \mathbb{R}^{d \times c}.$$

**Neural Feature Ansatz.**   In the above notation, the NFA (Radhakrishnan et al., 2024) posits that the neural feature matrix $W^\top W$ is proportional to the influence that the different subspaces of the input have on the output, which is captured by the Average Gradient Outer Product (AGOP) matrix. Namely, there is a power $s > 0$ such that

$$W^\top W \propto (\mathsf{AGOP})^s, \text{ where } \mathsf{AGOP} := \frac{1}{n}\sum_{i=1}^n (\nabla_h f_i)(\nabla_h f_i)^\top. \tag{NFA}$$

**Equivariant Neural Feature Ansatz.**   We will also compare to the eNFA proposed in Ziyin et al. (2025), which states

$$W^\top W \propto \mathsf{eNFA} := \frac{1}{n}\sum_{i=1}^n (\nabla_h \ell_i)(\nabla_h \ell_i)^\top. \tag{eNFA}$$

---

[1]More precisely, including the dependence on the parameters other than $W$, what this means is that we can partition the parameters as $\theta = [W, \theta_{-W}]$, and $f(x; \theta) = g(Wh(x; \theta_{-W}); x; \theta_{-W})$.

## 3 NEURAL FEATURES SATISFY FACT AT CONVERGENCE

In contrast to the empirically-derived NFA and eNFA, we seek to provide a relation derived from first principles. We proceed from the following simple observation: at a critical point of the loss, the features $h(x)$ are weighted by their influence on the final loss. This is stated in the following theorem.

**Theorem 3.1** (Features at Convergence Theorem). *If the parameters of the model are at a critical point of the loss with respect to $W$, then*

$$W^\top W = \mathsf{FACT} := -\frac{1}{n\lambda} \sum_{i=1}^n (\nabla_h \ell_i)(h(x_i))^\top . \tag{FACT}$$

*Proof.* The premise of the theorem implies that $\nabla_W \mathcal{L}_\lambda(\theta) = 0$, since $W$ is a subset of the model parameters. By left-multiplying by $W^\top$ and using the chain rule, we obtain

$$\begin{aligned}
0 &= W^\top (\nabla_W \mathcal{L}_\lambda(\theta)) \\
&= W^\top (\lambda W + \nabla_W \mathcal{L}(\theta)) \\
&= W^\top \left(\lambda W + \frac{1}{n} \sum_{i=1}^n \left( \frac{\partial \ell(g(\tilde{h}); y_i)}{\partial \tilde{h}} |_{\tilde{h}=Wh(x_i)} \right) h(x_i)^\top \right) \\
&= \lambda W^\top W + \frac{1}{n} \sum_{i=1}^n (\nabla_h \ell_i) h(x_i)^\top .
\end{aligned}$$

The theorem follows by rearranging and dividing by $\lambda$. $\qquad\square$

Theorem 3.1 is a straightforward modification of the stationarity conditions. Nevertheless, as we argue in the remainder of this paper, it is a useful quantity to consider when studying feature learning in neural networks, and it is especially fruitful when viewed as a first-principles counterpart to the NFA. Before proceeding with applications of (FACT), we provide a few remarks and empirical validation.

*Remark* 3.2 (Symmetrizations of FACT). While $W^\top W$ is p.s.d., the quantity FACT is only guaranteed to be p.s.d. at critical points of the loss. This means that we can get several other identities at convergence by algebraically modifying the right-hand side in a way that preserves the symmetries of the left-hand side. For instance, since $W^\top W = (W^\top W)^\top$, we may conclude that at the critical points of the loss $W^\top W = \mathsf{FACT}^\top$ also holds. Similarly, using that $W^\top W = \sqrt{(W^\top W)(W^\top W)^\top}$, we may also conclude that at critical points $W^\top W = \sqrt{\mathsf{FACT} \cdot \mathsf{FACT}^\top}$ also holds.

*Remark* 3.3 (Empirical validation on real-world data). In Figure 2, we verify FACT on 5-layer ReLU MLPs trained until convergence on MNIST (LeCun, 1998) and CIFAR-10 (Krizhevsky et al., 2009) with Mean Squared Error loss and weight decay $10^{-4}$. We find that, at convergence, the two sides of the (FACT) relation generally have higher Pearson correlation than those of the (NFA) and (eNFA) relations. For hyperparameter details, see Appendix A.

*Remark* 3.4 (Backward form). There is also an analogous "backward" version of this equation, (bFACT), derived and empirically validated in Appendix B, that yields information about the left singular vectors of $W$ rather than the right singular vectors. Letting $\nabla_{Wh} \ell_i$ denote the gradient of the loss with respect to the output of the layer at data point $x_i$, we have

$$WW^\top = \mathsf{bFACT} := -\frac{1}{n\lambda} \sum_{i=1}^n (Wh(x_i))(\nabla_{Wh} \ell_i)^\top . \tag{bFACT}$$

## 4 FACT CAPTURES FEATURE LEARNING PHENOMENA IN MANY OF THE SAME WAYS AS THE NFA CONJECTURE

Having validated the FACT, we now turn to applications. We show that our first-principles FACT captures many feature learning phenomena in the same ways that the empirically-driven NFA conjecture has been previously shown to do. First, we show that the FACT can be used to design

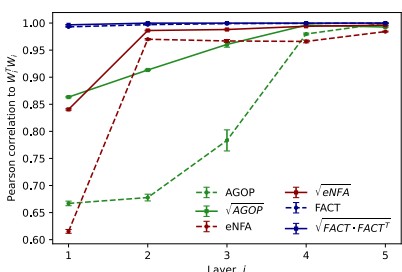
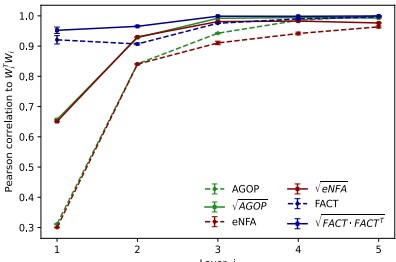

Figure 2: We train 5 hidden layer ReLU MLPs to interpolation (batch train loss $\leq 10^{-3}$) on MNIST and CIFAR-10. We plot Pearson correlation of FACT, AGOP, eNFA (with respect to each hidden layer input) to $W^T W$ for that layer. Curves are averaged over 5 independent runs. Both sides of the (FACT) are highly-correlated at convergence across layers. The power $s = 1/2$ for the NFA is suggested by prior work Radhakrishnan et al. (2024); Beaglehole et al. (2023); Mallinar et al. (2025).

learning algorithms that achieve high performance on tabular data based on adapting the recursive feature machine (RFM) algorithm of Radhakrishnan et al. (2024). We also show that this algorithm recovers important feature learning phenomena commonly studied in neural networks, such as phase transitions in sparse parity learning, and grokking of modular arithmetic.

## 4.1 BACKGROUND: RECURSIVE FEATURE MACHINES

The Recursive Feature Machine (RFM) algorithm (Radhakrishnan et al., 2024) builds upon classical kernel methods (Schölkopf, 2002), which rely on a kernel function $K(x, x')$ to measure data point similarity (e.g., Gaussian, Laplace). While kernel methods have been successful, they can be provably less sample-efficient than alternatives like neural networks that are able to learn features (Abbe et al., 2022; Damian et al., 2022).

To address these limitations, RFM learns a linear transformation $W \in \mathbb{R}^{d \times d}$ and applies a standard kernel $K$ to the transformed data: $K_W(x, x') = K(Wx, Wx')$. This learned $W$ enables RFM to identify salient features, akin to feature learning in a neural network layer (for example, if $W$ is low rank, its range contains the salient features while the orthogonal complement to its range contains the irrelevant features). Seeking to imitate the feature learning behavior in neural networks, Radhakrishnan et al. (2024) iteratively updates $W$ using a fixed-point iteration to satisfy the NFA condition. This is given in Algorithm 1, where the update equation on line 6 is given by

$$W_{t+1} \leftarrow (\mathsf{AGOP}_t)^{s/2}, \text{ where } \mathsf{AGOP}_t = \frac{1}{n} \sum_{i=1}^{n} (\nabla_x \hat{f}_t)(\nabla_x \hat{f}_t)^\top; \quad s > 0. \quad \text{(NFA-RFM update)}$$

---

**Algorithm 1** Recursive Feature Machine (based on NFA (Radhakrishnan et al., 2024) or FACT (ours))

1: **Input:** Training data $(X, y)$, kernel $K_W$, number of iterations $T$, ridge-regularization $\lambda \geq 0$
2: Initialize $W_0 \leftarrow I_{d \times d}$
3: **for** $t = 0$ **to** $T$ **do**
4:     Run kernel method: $\alpha_t \leftarrow (K_{W_t}(X, X) + n\lambda I)^{-1} y$
5:     Let $\hat{f}_t(x) := K_{W_t}(x, X)\alpha_t$ be the kernel predictor
6:     Update $W_t$, either with (NFA-RFM update) or (FACT-RFM update)
7: **end for**
8: **Output:** predictor $\hat{f}_T(x)$

---

| *Method* | FACT-RFM (no geom. averaging) | FACT-RFM (geom. averaging) | NFA-RFM | Kernel regression |
|---|---|---|---|---|
| *Accuracy (%)* | 85.22 | 84.99 | 85.10 | 83.71 |

Table 1: Average test accuracy over 120 datasets from the UCI corpus Fernandez-Delgado et al. (2014). We compare Laplace kernel regression with adaptively learned Laplace kernels using FACT and NFA, as well as no feature learning.

## 4.2 FACT-BASED RECURSIVE FEATURE MACHINES

We study RFM with a FACT-based update instead of an NFA-based update. Similarly to the above, let $\mathsf{FACT}_t$ be the FACT matrix corresponding to iteration $t$. We symmetrize in order to ensure that the update is p.s.d. Our FACT-based fixed-point iteration in line 6 of RFM is thus

$$W_{t+1} \leftarrow ((\mathsf{FACT}_t)(\mathsf{FACT}_t)^\top)^{1/4} . \qquad \text{(FACT-RFM update)}$$

We also study a variant of this update where we average geometrically with the previous iterate to ensure greater stability (which helps for the modular arithmetic task). This geometric averaging variant has the following update

$$W_{t+1} \leftarrow ((\mathsf{FACT}_t)(W_t^\top W_t)(W_t^\top W_t)(\mathsf{FACT}_t)^\top)^{1/8} . \qquad \text{(FACT-RFM update')}$$

The exponents in these updates are chosen so that the fixed points of these updates coincide with the FACT relation derived for networks at convergence in Theorem 3.1. See Appendix E for more details.

## 4.3 EXPERIMENTAL RESULTS COMPARING FACT-RFM TO NFA-RFM

We compare FACT-RFM to NFA-RFM across a range of settings (tabular datasets, sparse parities, and modular arithmetic).

**Tabular datasets.** The authors of Radhakrishnan et al. (2024) obtain state-of-the-art results using NFA-RFM on tabular benchmarks including that of Fernandez-Delgado et al. (2014) which utilizes 121 tabular datasets from the UCI repository. We run their same training and cross-validating procedure using FACT-RFM, and report results in Table 1. We find that FACT-RFM obtains roughly the same high accuracy performance as NFA-RFM. Both of these feature-learning methods improve over the next-best method found by Radhakrishnan et al. (2024), which is kernel regression with the Laplace kernel without any feature learning.

**Sparse parities.** We train FACT-RFM and NFA-RFM on the problem of learning sparse parities and find that both recover low-rank features. The problem of learning sparse parities has attracted attention with respect to feature learning dynamics of neural networks on multi-index models (Edelman et al., 2023; Abbe et al., 2023).

For training data we sample $n$ points in $d$-dimensions as $x \sim \{-\frac{1}{\sqrt{d}}, \frac{1}{\sqrt{d}}\}^d$. We experiment with sparsity levels of $k = 2, 3, 4$ by randomly sampling $k < d$ coordinate indices with which to construct our labels. Labels, $y$, are obtained from the product of the elements at each of the $k$ coordinates in the corresponding $x$ point and set to be 0 if the product is negative and 1 if the product is positive. We sample a held-out test set of 1000 points in the same manner.

We use the Mahalanobis Gaussian kernel in both FACT-RFM (with geometric averaging) and NFA-RFM with bandwidth 5 and train for 5 iterations. Our experiments use $d = 50$ and for $k = 1, 2$ we take $n = 500$, for $k = 3$ we take $n = 5000$, and for $k = 4$ we take $n = 50000$. The results of these experiments are given in Figure 3. We observe that both NFA-RFM and FACT-RFM learn this task and the features learned by both methods are remarkably similar and on the support of the sparse parity. Additionally, Figure 4 shows a phase transition in learning sparse parities when we take a smaller amount of data $n = 25000, k = 4$, which mimics the phase transition when training an MLP.

**Grokking modular arithmetic.** Mallinar et al. (2025) recently showed that NFA-RFM exhibits delayed generalization phenomena on modular arithmetic tasks, also referred to as "grokking". The

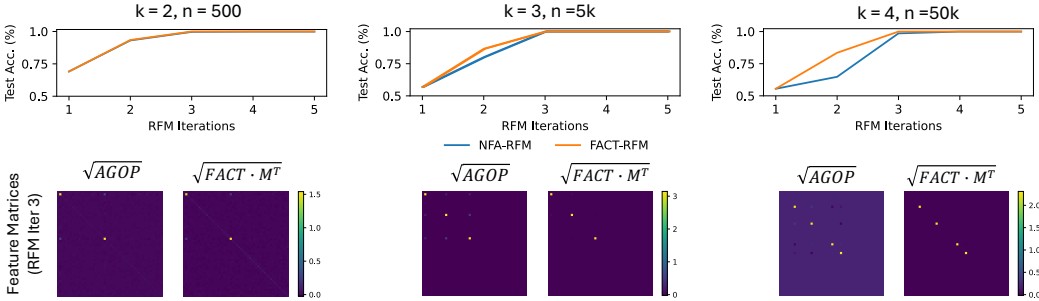

Figure 3: We train FACT-RFM and NFA-RFM using the Mahalanobis Gaussian kernel on sparse parity tasks. We train with $d = 50, k = 2, 3, 4$. The corresponding $\sqrt{\text{AGOP}}$ and $\sqrt{\text{FACT} \cdot M^\top}$ feature matrices are very similar and learn the support of the sparse parity.

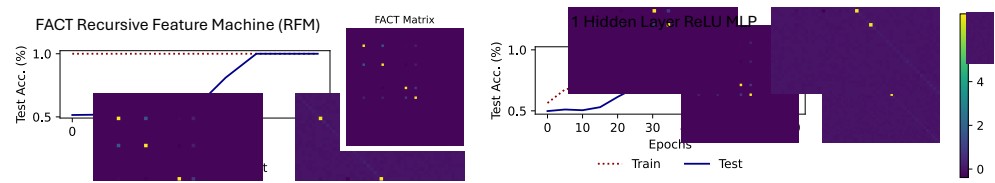

Figure 4: In the lower data regimes of $n = 25000$, $k = 4$, and $d = 50$, for sparse parity, the FACT-RFM algorithm reproduces phase transitions found in training neural networks.

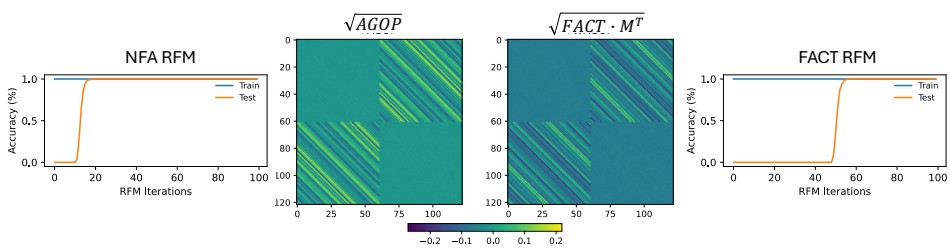

Figure 5: We train FACT-RFM and NFA-RFM on $(x + y) \mod 61$ for 75 iterations. Both methods achieve 100% test accuracy and exhibit delayed generalization aligned to the "grokking" phenomenon. We plot the square root of $\text{FACT} \cdot M^\top$ and AGOP and find that both methods learn block circulant feature transforms.

authors find that the square root of AGOP learns block circulant feature transformations on these problems. We train FACT-RFM (with geometric averaging) on the same modular arithmetic tasks and observe the same behavior. Figure 5 shows the square root of AGOP and FACT $\cdot M^\top$ after achieving 100% test accuracy on modular addition with modulus $p = 61$ when training on 50% of the data and testing on the other half. The feature matrices show block circulant structures.

## 5 COMPARISON OF NFA AND FACT FOR INNER-PRODUCT KERNELS

Having demonstrated that the first-principles FACT obtains many of the same feature learning phenomena as the empirically-conjectured NFA, it is natural to ask: is there a direct connection between these two relations? Does the FACT imply the NFA?

Our findings in this section suggest there is such a connection: the updates of NFA-RFM are proxies for the updates of FACT-RFM. Thus, the NFA-RFM algorithm can also be viewed as attempting to minimize the loss of the kernel method, regularized by the norm of the weights $\|W\|_F^2$. A similar claim was previously made in Gan & Poggio (2024), but the theoretical evidence provided was limited to the dynamics with one sample. Our analysis applies to training with more than one sample.

We restrict our analysis to inner-product kernels. The expressions for FACT and AGOP simplify considerably, as stated below. Below, we let $\alpha$ be first-order optimal dual weights for kernel regression with $\lambda$-ridge regularization computed in the RFM algorithm.

**Proposition 5.1** (Comparison of FACT and AGOP for inner-product kernels). *Suppose the kernel is an inner-product kernel of the form $K_W(x, x') = k(x^\top M x')$, where $M = W^\top W$. Then, we may write the* AGOP *and the* FACT *matrices explicitly as:*

$$\mathsf{AGOP} = \sum_{i,j=1}^{n} \tau(x_i, M, x_j) \cdot M x_i \alpha_i^\top \alpha_j x_j^\top M^\top \,,$$

$$\mathsf{FACT} \cdot M^\top = \sum_{i,j=1}^{n} k'(x_i^\top M x_j) \cdot M x_i \alpha_i^\top \alpha_j x_j^\top M^\top \,,$$

*where $\tau(x_i, M, x_j) := \frac{1}{n} \sum_{l=1}^{n} k'(x_l^\top M x_i) k'(x_l^\top M x_j)$.*

The proof is deferred to Appendix F. The proposition reveals that the matrix FACT $\cdot M^\top$ is positive semi-definite when the function $k$ is non-increasing (a condition satisfied by common choices like $k(t) = \exp(t)$ or $k(t) = t^2$). This property allows for a simplification of (FACT-RFM update'), which can be rewritten as a geometric average between the current feature matrix and the FACT term:

$$M_{t+1} \leftarrow (\mathsf{FACT}_t M_t)^{1/2} \,. \qquad \text{(FACT-RFM update' for inner-product kernels)}$$

This form should be compared with the NFA-RFM update, which also simplifies for inner-product kernels. We write the simplified form of the update below when the power $s$ is set to $1/2$:

$$M_t \leftarrow (\mathsf{AGOP})^{1/2} \,. \qquad \text{(NFA-RFM update for inner-product kernels)}$$

Notably, Proposition 5.1 also reveals that both updates share the same structural form. The difference lies in the specific factors involved: $\tau$ for the NFA update and $k'$ for the FACT update. Interestingly, both of these factors, $k'(x_i^\top M x_j)$ and $\tau(x_i, M, x_j)$, can be interpreted as measures of similarity between the data points $x_i$ and $x_j$. These measures increase when the transformed representations $W x_i$ and $W x_j$ are closer in the feature space, and decrease otherwise.

Consequently, if the similarity measures $\tau$ and $k'$ were approximately equal for most pairs of data points, this would explain the observed similarities in performance between the NFA-RFM and FACT-RFM methods, and account for their general agreement in tracking the feature learning process as it occurs in neural networks.

**Empirical validation.** We empirically validate the above explanation, showing that indeed $\tau(x_i, x_j, M)$ is approximately proportional to $k'(x_i^\top M x_j)$ for FACT-RFM in the challenging setting of arithmetic modulo $p = 61$ (where as demonstrated in Section 4.3 both algorithms converge to similar features). In Figure 6, we show that a best-fit line proportionally relating the two quantities achieves a good fit.

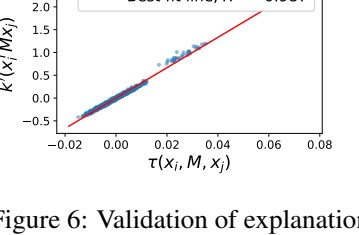

Figure 6: Validation of explanation for why AGOP and FACT are similar when FACT-RFM converges in the modular arithmetic task. Each point corresponds to a pair $(x_i, x_j)$ – we subsample 1000 points for visualization purposes.

## 6 NFA AND FACT MAY BE UNCORRELATED IN WORST-CASE SETTINGS

Finally, as a counterpoint to the analysis in the previous section, we show that when the data distribution is chosen adversarially, NFA and FACT can differ drastically even for shallow, two-layer nonlinear networks. Thus, FACT is perhaps a preferable alternative to the NFA.

We craft a dataset to maximimize their disagreement on a trained two-layer architecture $f(x; a, W) = a^\top \sigma(Wx)$ with quadratic activation $\sigma(t) = t^2$ and parameters $a \in \mathbb{R}^m$, $W \in \mathbb{R}^{m \times d}$ and any large enough width $m \geq 7$. For any $p \in (0, 1)$ and $\tau \in (0, 1)$, define the data distribution $\mathcal{D}(p, \tau)$ over $(x, y)$ such that $x$ is drawn from a mixture of distributions: $x \sim \text{Unif}[\{0, 1, 2\}^4]$ with probability $p$ and $x = (1, 1, 0, 0)$ with probability $1 - p$, and such that $y = f_*(x) = \tau x_1 x_2 + x_3 x_4 \in \mathbb{R}$. For appropriate choices of the hyperparameters, we show that the NFA prediction can be nearly uncorrelated with weights that minimize the loss, while the FACT provably holds.

**Theorem 6.1** (Separation between NFA and FACT in two-layer networks). *Fix any $s > 0$. For any $\epsilon \in (0, 1]$, there are hyperparameters $p_\epsilon, \tau_\epsilon, \lambda_\epsilon \in (0, 1)$ such that any parameters $\theta = (a, W)$ minimizing $\mathcal{L}_{\lambda_\epsilon}(\theta)$ on data distribution $\mathcal{D}(p_\epsilon, \tau_\epsilon)$ are nearly-uncorrelated with the* (NFA) *prediction:*

$$\text{corr}((\mathsf{AGOP})^s, W^\top W) < \epsilon,$$

*where the correlation* corr *is defined as* $\text{corr}(A, B) = \langle A, B \rangle / (\|A\|_F \|B\|_F)$. *In contrast,* (FACT) *holds because the weights are at a stationary point.*

**Proof intuition.** At a loss minimizer, the neural network approximates the true function $f_*$ because part of the data distribution is drawn from the uniform distribution. Therefore, since the neural network computes a quadratic because it has quadratic activations, one can show $\mathsf{AGOP} \approx \mathbb{E}_{x \sim \mathcal{D}(p_\epsilon, \tau_\epsilon)}[(\nabla_x f_*)(\nabla_x f_*)^\top] \approx \tau_\epsilon^2 (e_1 + e_2)(e_1 + e_2)^\top + O(p_\epsilon)$, For small $p_\epsilon$, this matrix has most of its mass in the first two rows and first two columns.

On the other hand, the weight decay in training the neural network means that at convergence the norm of the network weights is minimized given the function it computes. Since the neural network approximates the true function $f_*$, in order to minimize the total norm of the weights, $W^\top W$ must have most of its mass on the last two rows and columns when $\tau_\epsilon$ is small. This is in contrast to AGOP, since as we have argued that has most of its mass on the first two rows and columns. Thus, the NFA prediction is not met. On the other hand, the FACT prediction is provably met by Theorem 3.1. The formal proof is in Appendix D.

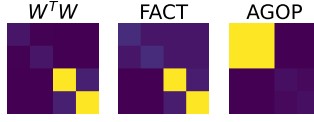

Figure 7: The FACT and NFA are uncorrelated at convergence on the synthetic dataset.

The construction is empirically validated in Figure 7, which is the result of training a width-10 network for $10^6$ iterations of Adam with learning rate $0.01$ on the population loss with $\tau = 0.02$, $p = 10^{-5}$, $\lambda = 10^{-5}$. At convergence, FACT achieves $0.994$ cosine similarity with $W^\top W$, while AGOP achieves $< 0.068$ cosine similarity.

# 7 DISCUSSION

This work pursues a first principles approach to understanding feature learning by deriving a condition that must hold in neural networks at critical points of the train loss. Perhaps the most striking aspect of our results is that FACT is based only on **local optimality** conditions of the loss. Nevertheless, in Section 4.3 we show that when used to drive the RFM algorithm, FACT recovers interesting **global behaviors** of neural networks: including high-performing feature learning for tabular dataset tasks, and grokking and phase transition behaviors on arithmetic and sparse parities datasets.

The usefulness of FACT is especially surprising since there is no reason for FACT to be correlated to neural feature matrices during most of training, prior to interpolating the train loss; and indeed FACT does have low correlation for most epochs (although $\sqrt{\text{FACT} \cdot \text{FACT}^\top}$ has nontrivial correlation), before sharply increasing to near-perfect correlation; see Figure 8. This is a potential limitation to using FACT to understand the evolution of features *during training*, rather than in the terminal phase. Therefore, it is of interest to theoretically derive a quantity with more stable correlation over training.

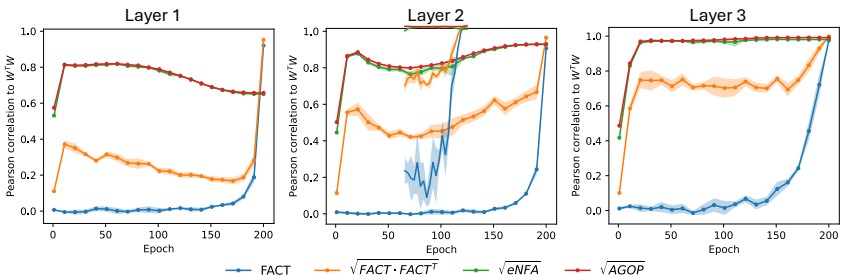

Figure 8: We train 5 layer ReLU MLPs to interpolation on CIFAR-10 and plot Pearson correlation vs. epochs comparing FACT, AGOP, eNFA to neural feature matrices for the first three layers of the model. Curves are averaged over five independent runs.

An additional limitation is that there are data distributions, such as sparse parity, where FACT-RFM becomes unstable if continued iterations are performed after convergence, so early stopping is necessary. Understanding this phenomenon may help derive relations that improve over FACT.

Finally, our formulation of FACT for neural networks requires non-zero weight decay. This is a reasonable assumption for real-world neural network training (LLMs are often pretrained with a reasonably large weight decay factor), but raises othe question of whether it is possible to compute FACT in the zero weight-decay limit. In Section 5, we formulate FACT in a way that only relies on the kernel dual weights, the learned features, and the data. Therefore it may be possible to compute FACT in neural networks through the network's empirical neural tangent kernel (Jacot et al., 2018; Long, 2021), which would allow using FACT without requiring weight decay.

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

## A  HYPERPARAMETERS IN EMPIRICAL VALIDATION OF FACT AND FURTHER EXPERIMENTS

In the empirical validation of FACT in Figure 2, we train the networks until convergence, which we operationalize as the point at which batch train loss $\leq 10^{-3}$ is achieved. The results are for training 5-layer fully-connected networks with Mean-Squared Error (MSE) loss for 200 epochs using SGD, momentum 0.9, initial learning rate $10^{-1}$, cosine decay learning rate schedule, weight decay $10^{-4}$, batch size 64, depth 3, and hidden width 1024 for MNIST and 3072 for CIFAR-10, and standard PyTorch initialization.

### A.1  VARYING THE NUMBER OF DATA POINTS AND THE WEIGHT DECAY

We experiment in the above setting with varying the number of training data points, as well as the weight decay. Table 2 reports the Pearson correlation to $W_1^\top W_1$ after training for the square root of AGOP, the square root of the eNFA (Ziyin et al., 2025), and the FACT. In addition we report the correlation to a symmetrization of FACT given that unlike the other quantities the FACT is only guaranteed to be p.s.d. at exact convergence and we only train to approximate convergence with batch train loss $\leq 10^{-3}$. In these experiments, the eNFA and NFA predictions without square root are both less correlated to the $W_1^\top W_1$ than their square root counterparts. In all cases except for MNIST $n = 1000$ we find that FACT and its symmetrization are better correlated to $W_1^\top W_1$ than both AGOP and eNFA and their square roots.

| | Pearson Correlation to $W^\top W$ after training | | | | | |
| --- | --- | --- | --- | --- | --- | --- |
| | MNIST | | | | CIFAR-10 ($n = 50K$) | |
| | $n = 1K$ | $n = 2K$ | $n = 5K$ | $n = 60K$ | wt decay = $10^{-3}$ | wt decay = $10^{-4}$ |
| FACT | 0.890 | 0.988 | 0.995 | 0.999 | 0.830 | 0.896 |
| $\sqrt{\text{FACT} \cdot \text{FACT}^\top}$ | **0.956** | **0.998** | **1.000** | **1.000** | **0.929** | **0.943** |
| $\sqrt{\text{AGOP}}$ (NFA with $s = 1/2$) | 0.899 | 0.923 | 0.924 | 0.909 | 0.793 | 0.722 |
| $\sqrt{\text{eNFA}}$ | 0.903 | 0.923 | 0.917 | 0.903 | 0.797 | 0.714 |

Table 2: We report the correlation of the Neural Feature Matrix $W^\top W$ to the corresponding feature matrices at interpolation, given by the square root of AGOP, the equivariant NFA, the FACT and a symmetrization of FACT. Here $W$ is the first layer of the network. The weight decay for MNIST-trained networks is $10^{-4}$, but we experiment with weight decays $10^{-3}$ and $10^{-4}$ for CIFAR-trained networks. For MNIST, we experiment with changing the training set size as well.

## B  BACKWARD FORM OF FACT

We provide here an analogous "backward" form of the FACT condition, which applies to $WW^\top$ instead of $W$.

Recall from Section 2 that the neural network depends on $W$ as

$$f(x) = g(Wh(x), x).$$

Out of convenience, we introduce notation to denote the gradient of the loss **with respect to the output of the layer** at data point $x_i$. We write

$$\nabla_{Wh}\ell_i := \frac{\partial \ell(g(\tilde{h}, x); y_i)}{\partial \tilde{h}} \Big|_{\tilde{h} = Wh(x_i)} \in \mathbb{R}^d.$$

With this notation in hand, the backward form of the FACT, which gives information about the left singular vectors, is:

**Theorem B.1.** *If the parameters of the network are at a differentiable, critical point of the loss with respect to $W$, then*

$$WW^\top = \text{bFACT} := -\frac{1}{n\lambda} \sum_{i=1}^{n} (Wh(x_i))(\nabla_{Wh}\ell_i)^\top. \quad \text{(bFACT)}$$

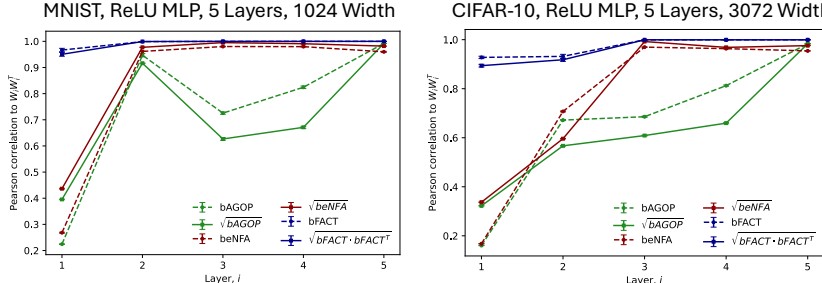

Figure 9: We train 5 hidden layer ReLU MLPs to interpolation on MNIST and CIFAR-10. We plot the Pearson correlation of the backward versions of FACT, AGOP, eNFA (with respect to pre-activation outputs of a layer) and compare to $WW^T$ for that layer. Curves are averaged over 5 independent runs.

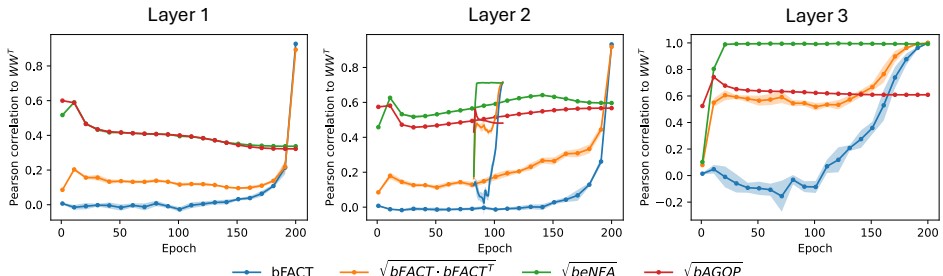

Figure 10: We train 5 hidden layer ReLU MLPs to interpolation on CIFAR-10. We plot the Pearson correlation of the backward versions of FACT, AGOP, eNFA (with respect to pre-activation outputs of a layer) vs. epochs. Curves are averaged over 5 independent runs.

*Proof.* Left-multiplying by $W$, we obtain

$$
\begin{aligned}
0 &= W(\nabla_W \mathcal{L}_\lambda(\theta))^\top \\
&= W(\lambda W + \nabla_W \mathcal{L}(\theta))^\top \\
&= W(\lambda W + \frac{1}{n}\sum_{i=1}^{n}(\frac{\partial \ell(g(\tilde{h}); y_i)}{\partial \tilde{h}}\big|_{\tilde{h}=Wh(x_i)} h(x_i)^\top)^\top \\
&= \lambda WW^\top + (Wh(x_i))(\nabla_{Wh}\ell_i)^\top
\end{aligned}
$$

Rearranging and dividing by $\lambda$ proves (bFACT). $\qquad\square$

Again, we may symmetrize both sides of the equation and still get valid equations that hold at critical points of the loss: for instance we have $WW^\top = \sqrt{\text{bFACT} \cdot \text{bFACT}^\top}$.

In the same way that we compute bFACT, we can compute an analogous "backward" version of AGOP which is given by,

$$
\text{bAGOP} = \frac{1}{n}\sum_{i=1}^{n}(\nabla_{Wh}f_i)(\nabla_{Wh}f_i)^\top
$$

and consider whether this models the left singular vectors of layer weights as well. The backward eNFA is as computed in (Ziyin et al., 2025). Figures 9 and 10 show the complete set of comparisons for backward versions of FACT, AGOP, eNFA to their respective neural feature matrices compared across both depth and epochs. The hyperparameters and training setup are the same as that described in Appendix A.

## C  CASE STUDY FOR DEEP LINEAR NETWORKS

In this appendix, we compare the predictions of FACT and NFA in the toy setting of deep linear networks, which have received significant attention in the theoretical literature as a simplified setting for studying training dynamics Arora et al. (2019a); Ziyin et al. (2022); Marion & Chizat (2024); Saxe et al. (2014); Arora et al. (2018). A deep linear network $f : \mathbb{R}^d \to \mathbb{R}^c$ is parameterized as

$$f(x) = W_L \cdot W_{L-1} \cdots W_1 x$$

for $W_1 \in \mathbb{R}^{h \times d}, W_2, \ldots, W_{L-1} \in \mathbb{R}^{h \times h}, W_L \in \mathbb{R}^{c \times h}$. We fit the network on data points $x \sim \mathcal{N}(0, I_d)$ and labels given by a ground truth linear transformation $f^*(x) = W^* x$ where $W^* \in \mathbb{R}^{c \times d}$.

In this setting, Radhakrishnan et al. (2025) show that the exponent $s$ in (NFA) must scale as $1/L$ in order for the NFA prediction to be correct. Thus, unlike the FACT, the NFA has a tunable hyperparameter that must depend on the particular architecture involved. We rederive this dependence of the exponent on the architecture for completeness.

**Informal derivation of NFA power dependence on depth**  In this setting, the NFA prediction for the first layer can be computed as

$$\mathsf{AGOP} = W_1^\top \cdots W_{L-1}^\top W_L^\top W_L \cdot W_{L-1} \cdots W_1 \,,$$

So, when training has converged and the network is close to fitting the ground truth $W^*$, we have

$$\mathsf{AGOP} \approx (W^*)^\top W^* \,.$$

It is known that weight decay biases the solutions of deep linear networks to be "balanced" at convergence Gunasekar et al. (2017); Arora et al. (2019b), meaning that the singular values at each layer are equal. When the layers are balanced we should therefore heuristically expect that, after training, we have

$$W_1^\top W_1 \approx ((W^*)^\top W^*)^{1/L} \,,$$

because singular values multiply across the $L$ layers. Putting the above equations together, at convergence we have

$$W_1^\top W_1 \approx (\mathsf{AGOP})^{1/L} \,.$$

For $L = 2$, we recover the prescription of using $\sqrt{\mathsf{AGOP}}$ suggested by Radhakrishnan et al. (2024); Beaglehole et al. (2023); Mallinar et al. (2025). However when $L \neq 2$, this is no longer the best power. Our analysis suggests that the AGOP power must be tuned with the depth of the network – on the other hand, FACT does not need this tunable parameter.

We empirically validate this in Figure 11, with deep linear networks with $d = 10, c = 5, h = 512$ and varying the depth $L$, and sample $W^* \in \mathbb{R}^{c \times d}$ with independent standard Gaussian entries. The train dataset is of size $n = 5000$ where $x_i \sim \mathcal{N}(0, I_d)$. We train to convergence using the Mean Squared Error (MSE) loss for 5000 epochs with SGD, minibatch size 128, learning rate of $5 \times 10^{-3}$, weight decay of $10^{-2}$, and standard PyTorch initialization. After training, the singular values of all of the weight matrices are identical after training, indicating balancedness has been achieved.

## D  PROOF OF THEOREM 6.1, SEPARATING FACT AND NFA FOR TWO-LAYER NETWORKS

We provide the proof of Theorem 6.1, restating the theorem as Theorem D.1 for convenience.

**Setup**  Consider a trained two-layer architecture $f(x; a, W) = a^\top \sigma(Wx)$ with quadratic activation $\sigma(t) = t^2$ and parameters $a \in \mathbb{R}^m, W \in \mathbb{R}^{m \times d}$. For any $p \in (0, 1)$ and $\tau \in (0, 1)$, define the data distribution $\mathcal{D}(p, \tau)$ over $(x, y)$ such that $x$ is drawn from a mixture of distributions: $x \sim \mathrm{Unif}[\{0, 1, 2\}^4]$ with probability $p$ and $x = (1, 1, 0, 0)$ with probability $1 - p$, and such that $y = f_*(x) = \tau x_1 x_2 + x_3 x_4 \in \mathbb{R}$.

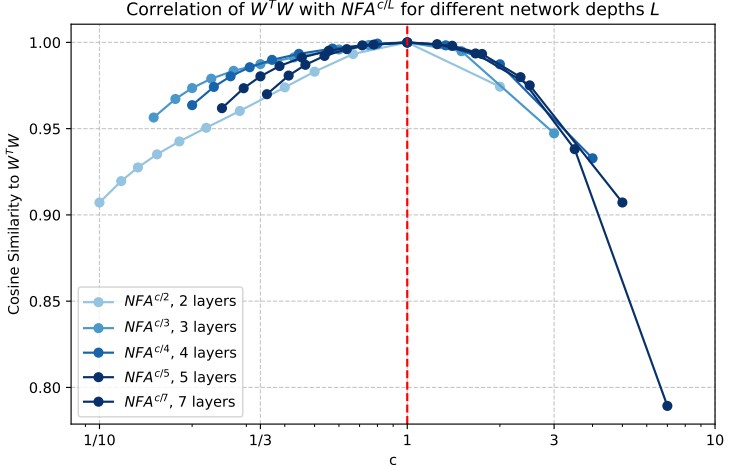

Figure 11: Deep $L$-layer linear networks trained to convergence on synthetic data. $\mathsf{AGOP}^{1/L}$ has cosine similarity close to 1 to the NFM ($W_1^\top W_1$), which validates the derivation in Appendix C. For all of these network depths, FACT has cosine similarity $\geq 0.999$, and there are no tunable hyperparameters that depend on depth.

**Theorem D.1** (Separation between NFA and FACT in two-layer networks; restated Theorem 6.1).
*Fix $s > 0$ to be the NFA power. For any $\epsilon \in (0,1)$, there are hyperparameters $p_\epsilon, \tau_\epsilon, \lambda_\epsilon \in (0,1)$ such that any parameters $\theta = (a, W)$ minimizing $\mathcal{L}_{\lambda_\epsilon}(\theta)$ on data distribution $\mathcal{D}_\epsilon := \mathcal{D}(p_\epsilon, \tau_\epsilon)$ are nearly-uncorrelated with the NFA prediction:*

$$\mathrm{corr}((\mathsf{AGOP})^s, W^\top W) < \epsilon\,,$$

*where $\mathrm{corr}(A, B) = \langle A, B\rangle/(\|A\|_F \|B\|_F)$ is the correlation. On the other hand, the FACT prediction holds.*

*Proof.* Set $\tau_\epsilon = \epsilon^3$, $p_\epsilon = \epsilon^8$, $\lambda_\epsilon = \epsilon^{32} p_\epsilon$. The outline of the proof that the loss-minimizing weights and AGOP are uncorrelated is to first show that there is a set of weights $\bar{a}, \bar{W}$ such that the loss $\mathcal{L}_{\lambda_\epsilon}(\bar{a}, \bar{W})$ is small. This implies that at any minimizer $a^*, W^*$ we must also have that $\mathcal{L}_{\lambda_\epsilon}(a^*, W^*)$ is small. In turn, this means that the estimated function $\hat{f}(\cdot) = f(\cdot; a^*, W^*)$ must be close to the true function $f_*(x) = \tau_\epsilon x_1 x_2 + x_3 x_4$. Finally, this will let us compare the AGOP to the loss-minimizing weights $\hat{a}, \hat{W}$.

1. Construct weights with low loss. Construct $\bar{W} = [\bar{w}_1, \ldots, \bar{w}_m]^\top \in \mathbb{R}^{m \times d}$ and $\bar{a} = [\bar{a}_1, \ldots, \bar{a}_m]^\top \in \mathbb{R}^{m \times 1}$ by letting $\bar{w}_1 = e_1 + e_2$, $\bar{w}_2 = e_1$, $\bar{w}_3 = e_2$, $\bar{w}_4 = e_3 + e_4$, $\bar{w}_5 = e_3$, $\bar{w}_6 = e_4$, $\bar{a}_1 = \tau_\epsilon/2$, $\bar{a}_4 = 1/2$, $\bar{a}_2 = \bar{a}_3 = -\tau_\epsilon$, $\bar{a}_5 = \bar{a}_6 = -1$, and $\bar{w}_j = 0$ and $\bar{a}_j = 0$ for all $j \geq 7$. One can check that $f(x; \bar{a}, \bar{W}) = f_*(x)$ for all $x$, and that $\|\bar{W}\|_F^2 + \|a\|^2 \leq 13$. Therefore

$$\mathcal{L}_{\lambda_\epsilon}(\hat{a}, \hat{W}) \leq \mathcal{L}_{\lambda_\epsilon}(\bar{a}, \bar{W}) \leq 169\lambda_\epsilon\,. \tag{D.1}$$

2. Conclude that $\hat{f} \approx f_*$. Define $\hat{f}(\cdot) = f(\cdot; \hat{a}, \hat{W})$. Since $\hat{f}$ and $f_*$ are homogeneous quadratic functions, we may write them as

$$\hat{f}(x) = \sum_{1 \leq i \leq j \leq 4} \hat{c}_{ij} x_i x_j \quad \text{and} \quad f_*(x) = \sum_{1 \leq i \leq j \leq 4} c_{ij} x_i x_j\,.$$

Let us show that the coefficients $\{c_{ij}\}$ must be close to the estimated coefficients $\{\hat{c}_{ij}\}$ using a Fourier-analytic calculation. Define the distribution $\mathcal{U} = \mathrm{Unif}[\{0, 1, 2\}^4]$ and the inner product between a pair of functions $\langle g, h\rangle_{\mathcal{U}} = \mathbb{E}_{x \sim \mathcal{U}}[g(x)h(x)]$. Also define the functions

$$\chi^{(0)}(t) = \begin{cases} 3, & t = 0 \\ 0, & t \in \{1,2\} \end{cases}, \quad \chi^{(1)}(t) = \begin{cases} -4.5, & t = 0 \\ 6, & t = 1 \\ -1.5 & t = 2 \end{cases}, \quad \chi^{(2)}(t) = \begin{cases} 1.5, & t = 0 \\ -3, & t = 1 \\ 1.5, & t = 2 \end{cases}$$

and for any vector of degrees $\alpha \in \{0,1,2\}^k$ define $\chi_\alpha : \{0,1,2\}^4 \to \mathbb{R}$ by $\chi_\alpha(x) = \prod_{i=1}^k \chi^{(\alpha_i)}(x_i)$. These functions have been picked so that for any $\alpha' \in \{0,1,2\}^k$ and monomial $h_{\alpha'}(x) = x_1^{\alpha'_1} x_2^{\alpha'_2} \dots x_k^{\alpha'_k}$, we have $\langle h_\alpha, \chi_{\alpha'} \rangle_{\mathcal{U}} = 1(\alpha = \alpha')$. Therefore, for any $1 \le i \le j \le 4$, there is $\alpha \in \{0,1,2\}$ such that

$$c_{ij} = \langle f_*, \chi_\alpha \rangle_{\mathcal{U}} \text{ and } \hat{c}_{ij} = \langle \hat{f}, \chi_\alpha \rangle_{\mathcal{U}}.$$

Therefore, by Cauchy-Schwarz, for any $1 \le i \le j \le 4$ and corresponding $\alpha$ we have

$$|c_{ij} - \hat{c}_{ij}| = |\langle f_* - \hat{f}, \chi_\alpha \rangle_{\mathcal{U}}| \le \|f_* - \hat{f}\|_{\mathcal{U}} \|\chi_\alpha\|_{\mathcal{U}} \le 6^4 \|f_* - \hat{f}\|_{\mathcal{U}}$$

Now we can apply our previous bound in (D.1), which implies that $\mathbb{E}_{(x,y) \sim \mathcal{D}_\epsilon}[(\hat{f}(x) - f_*(x))^2] \le \mathcal{L}(\hat{a}, \hat{W}) \le 169\lambda_\epsilon$, and in turn means that

$$\|f - f_*\|_{\mathcal{U}}^2 = \langle f - f^*, f - f^* \rangle_{\mathcal{U}} = \mathbb{E}_{x \sim \text{Unif}[\{0,1,2\}^4]}[(\hat{f}(x) - f_*(x))^2] \le 169\lambda_\epsilon/p_\epsilon.$$

So the estimated coefficients $\{\hat{c}_{ij}\}$ are close to the true coefficients $\{c_{ij}\}_{ij}$, i.e., for any $1 \le i \le j \le 4$,

$$|c_{ij} - \hat{c}_{ij}| \le 17000\sqrt{\lambda_\epsilon/p_\epsilon} := \delta_\epsilon. \tag{D.2}$$

Notice that $\delta_\epsilon \le 17000\epsilon^{16} \le 1/10$ for small enough $\epsilon$.

3a. Estimate the AGOP of $\hat{f}$. Since we have shown $\hat{f} \approx f$, the AGOP of the estimated function can be well approximated as follows.

$$\text{AGOP}(\hat{f}, \mathcal{D}_\epsilon) = \mathbb{E}_{(x,y) \sim \mathcal{D}_\epsilon}\left[\frac{\partial \hat{f}}{\partial x} \frac{\partial \hat{f}}{\partial x}^\top\right]$$

$$= (1 - p_\epsilon)\frac{\partial \hat{f}}{\partial x} \frac{\partial \hat{f}}{\partial x}^\top \Big|_{x=(1,1,0,0)} + p_\epsilon \mathbb{E}_{(x,y) \sim \mathcal{U}}\left[\frac{\partial \hat{f}}{\partial x} \frac{\partial \hat{f}}{\partial x}^\top\right]$$

Since $|\hat{c}_{ij}| \le |c_{ij}| + \delta_\epsilon \le 1 + \delta_\epsilon \le 11/10$ for all $i,j$ it must hold that $\|\frac{\partial \hat{f}}{\partial x} \frac{\partial \hat{f}}{\partial x}^\top\|_F \le 100$ for all $x \in \{0,1,2\}^4$, so

$$\left\|\text{AGOP}(\hat{f}, \mathcal{D}_\epsilon) - \frac{\partial \hat{f}}{\partial x} \frac{\partial \hat{f}}{\partial x}^\top \Big|_{x=(1,1,0,0)}\right\|_F \le 100 p_\epsilon \tag{D.3}$$

Finally, $\frac{\partial \hat{f}}{\partial x}\Big|_{x=(1,1,0,0)} = 2\hat{c}_{11}e_1 + 2\hat{c}_{22}e_2 + \hat{c}_{12}(e_1 + e_2) + (\hat{c}_{13} + \hat{c}_{23})e_3 + (\hat{c}_{14} + \hat{c}_{24})e_4$, and $c_{11} = c_{22} = c_{13} = c_{14} = c_{23} = c_{24} = 0$ and $c_{12} = \tau_\epsilon$, which means that

$$\left\|\frac{\partial \hat{f}}{\partial x} \frac{\partial \hat{f}}{\partial x}^\top \Big|_{x=(1,1,0,0)} - \tau_\epsilon^2 (e_1 + e_2)(e_1 + e_2)^\top\right\|_F \le 20\left\|\frac{\partial \hat{f}}{\partial x}\Big|_{x=(1,1,0,0)} - \tau_\epsilon(e_1 + e_2)\right\|_F \le 1000\sqrt{\delta_\epsilon}. \tag{D.4}$$

Putting the (D.3) and (D.4) together with the triangle inequality, we conclude our estimate of the AGOP

$$\|\text{AGOP}(\hat{f}, \mathcal{D}_\epsilon) - \tau_\epsilon^2 (e_1 + e_2)(e_1 + e_2)^\top\|_F \le 100 p_\epsilon + 1000\sqrt{\delta_\epsilon}. \tag{D.5}$$

3a. Estimate powers of the AGOP of $\hat{f}$. Next, let $s > 0$ be the power of the AGOP that we will take.

Let $\lambda_1 \ge \dots \ge \lambda_4 \ge 0$ be the eigenvalues of AGOP, with a corresponding set of orthonormal eigenvectors $v_1, \dots, v_4 \in \mathbb{R}^4$. By Weyl's inequality, since $\tau_\epsilon^2 \ge 100 p_\epsilon + 1000\sqrt{\delta_\epsilon}$ for small enough $\epsilon$,

$$\lambda_1 \ge 2\tau_\epsilon^2 - (100 p_\epsilon + 1000\sqrt{\delta_\epsilon}) \gtrsim \epsilon^6$$

and

$$\lambda_1 \lesssim \epsilon^6$$

and

$$0 \leq \lambda_4 \leq \lambda_3 \leq \lambda_2 \leq 100 p_\epsilon + 1000 \sqrt{\delta_\epsilon} \lesssim \epsilon^8 \,.$$

Additionally, let $P_\perp$ be the projection to the orthogonal subspace spanned by $\{(e_1 + e_2)\}$. By the Davis-Kahan $\sin(\Theta)$ theorem Davis & Kahan (1970),

$$\|P_\perp v_1\| \leq \frac{100 p_\epsilon + 1000 \sqrt{\delta_\epsilon}}{\tau_\epsilon^2} \lesssim \epsilon^2.$$

Notice that $\mathsf{AGOP} = \sum_{i=1}^{4} \lambda_i^s v_i v_i^\top$, which we will use later.

4. Estimate the loss-minimizing weights. Now let us estimate the loss-minimizing weights, $\hat{a}, \hat{W}$. The argument here is split into two parts: we want to (a) show that $\hat{W}$ is small in the first and second columns, and (b) show that $\hat{W}$ is large in the third or fourth column. These two facts combined will be enough show that $\hat{W}^\top \hat{W}$ is close to uncorrelated to the AGOP.

4a. Show that $\hat{W}_{1:m,1}$ and $\hat{W}_{1:m,2}$ are small. Define weights $a', W'$ by letting $a' = \hat{a}$ and $W' = \begin{bmatrix} 0 & 0 & \hat{W}_{1:m,3} & \hat{W}_{1:m,4} \end{bmatrix}$. In other words, we have zeroed out the coefficients of the variables $x_1$ and $x_2$ in the first layer. Then define

$$f'(x) = (a')^\top \sigma((W')x) \,.$$

If we write $f'(x) = \sum_{1 \leq i \leq j \leq 4} = c'_{ij} x_i x_j$, notice that $c'_{11} = c'_{12} = c'_{13} = c'_{14} = c'_{23} = c'_{24} = 0$ and that $c'_{34} = \hat{c}_{34}$, $c'_{33} = \hat{c}_{33}$, and $c'_{44} = \hat{c}_{44}$. Now, let $a'', W''$ be weights minimizing $\|a''\|^2 + \|W''\|_F^2$ such that

$$f'(\cdot) \equiv f(\cdot; a'', W'') \,.$$

By the construction in Lemma D.2, we may assume without loss of generality that all but 4 neurons are nonzero: i.e., that $a''_5 = \cdots = a''_m = 0$ and $W''_{5,1:4} = \ldots W''_{m,1:4} = 0$. Now the difference between the network after the zeroing out and the current network is $\hat{f}(x) - f'(x) = \sum_{1 \leq i \leq j \leq 4} \tilde{c}_{ij} x_i x_j$ where

$$|\tilde{c}_{ij}| = |\hat{c}_{ij} - c'_{ij}| \leq \tau_\epsilon + \delta_\epsilon \,.$$

So by Lemma D.2, this difference can be represented on four neurons with a cost of at most $100(\tau_\epsilon + \delta_\epsilon)^{2/3}$. Therefore, by editing the weights $a'', W''$ we can construct weights $a''', W'''$ such that $\hat{f}(\cdot) \equiv f(a''', W''')$ and

$$\begin{aligned}
\|\hat{a}\|^2 + \|\hat{W}\|_F^2 &\leq \|a'''\|^2 + \|W'''\|_F^2 \\
&\leq \|a''\|^2 + \|W''\|_F^2 + 100(\tau_\epsilon + \delta_\epsilon)^{2/3} \\
&= \|\hat{a}\|^2 + \|\hat{W}\|_F^2 - \|\hat{W}_{1:m,1}\|^2 - \|\hat{W}_{1:m,2}\|^2 + 100(\tau_\epsilon + \delta_\epsilon)^{2/3} \,.
\end{aligned}$$

So we can conclude that the norm of the weights in the first and second column is small

$$\|\hat{W}_{1:m,1}\|^2 + \|\hat{W}_{1:m,2}\|^2 \leq 100(\tau_\epsilon + \delta_\epsilon)^{2/3} \lesssim \epsilon^2 \,. \tag{D.6}$$

4b. Show that at least one of $\hat{W}_{1:m,3}$ or $\hat{W}_{1:m,4}$ is large. Finally, let us show that either the third or fourth column of the weights is large.

$$\begin{aligned}
0.9 \leq c_{34} - \delta_\epsilon \leq \hat{c}_{34} &\leq \sum_{i=1}^{m} \hat{a}_i \hat{W}_{i,3} \hat{W}_{i,4} \\
&\leq \|\hat{a}\| \sqrt{\sum_{i=1}^{m} (\hat{W}_{i,3} \hat{W}_{i,4})^2} \\
&\leq \|\hat{a}\| \sqrt{\sum_{i=1}^{m} (\hat{W}_{i,3})^2} \sqrt{\sum_{i=1}^{m} \hat{W}_{i,4}^2} \\
&= \|\hat{a}\| \|\hat{W}_{1:m,3}\| \|\hat{W}_{1:m,4}\| \,.
\end{aligned}$$

From the construction of the weights $\bar{a}, \bar{W}$ in the first step of this proof, we know that $\|\hat{a}\|^2 \le \|\bar{a}\|^2 + \|\bar{W}\|^2 \le 13$. So $\|\hat{a}\| \le 4$. We conclude that

$$\max(\|\hat{W}_{1:m,3}\|, \|\hat{W}_{1:m,4}\|) \ge 1/3. \tag{D.7}$$

5. Compare AGOP to loss-minimizing weights. Finally, let us compare the NFA approximation (D.5) to the facts proved in (D.6) and (D.7) about the loss-minimizing weights. From (D.5) and (D.6) and $\|\hat{W}\|_F^2 \le 13$ and the calculations in step 3b, we conclude that

$$\langle (\mathsf{AGOP}(\hat{f}, \mathcal{D}_\epsilon))^s, \hat{W}^\top \hat{W} \rangle = \sum_{i=1}^{4} \lambda_i^s \langle v_i v_i^\top, \hat{W}^\top \hat{W} \rangle$$
$$\lesssim (\|\hat{W}_{1:m,1}\|^2 + \|\hat{W}_{1:m,2}\|^2)(\lambda_1^s) + \lambda_1^s \|P_\perp v_1\|^2 + \lambda_2^s + \lambda_3^s + \lambda_4^s$$
$$\lesssim \epsilon^2 \epsilon^{6s} + \epsilon^{8s}.$$

From (D.5) and step 3b we conclude that

$$\|(\mathsf{AGOP}(\hat{f}, \mathcal{D}_\epsilon))^s\|_F \gtrsim (2\tau_\epsilon^2 - 100 p_\epsilon - 1000\sqrt{\delta_\epsilon})^s \ge \tau_\epsilon^{2s} \gtrsim \epsilon^{6s}.$$

From (D.7), we conclude that

$$\|\hat{W}^\top \hat{W}\| \ge 1/9 \gtrsim 1.$$

which implies that

$$\mathsf{corr}(\mathsf{AGOP}(\hat{f}, \mathcal{D}_\epsilon), \hat{W}^\top \hat{W}) \lesssim (\epsilon^2 \epsilon^{6s} + \epsilon^{8s})/\epsilon^{6s} \lesssim \epsilon^{2s} + \epsilon^2,$$

which can be taken arbitrarily small by sending $\epsilon$ to 0. $\qquad \square$

The Lemma that we used in the proof of this theorem is below.

**Lemma D.2** (The minimum-norm weight solution for a network with quadratic activation). *Let $f(x; a, W) = a^\top \sigma(Wx)$ be a neural network with quadratic activation function $\sigma(t) = t^2$ and weights $W \in \mathbb{R}^{m \times d}, a \in \mathbb{R}^m$ for $m \ge d$. Then, for any homogeneous quadratic function $f(x) = x^\top Q x$, where $Q = Q^\top$, the minimum-norm neural network that represents $f$ has cost:*

$$2\sum_{i=1}^{d} \sigma_i(Q)^{2/3} = \min_{a, W}\{\|a\|^2 + \|W\|_F^2 : f(\cdot; a, W) \equiv f(\cdot)\},$$

*and this can be achieved with a network that has at most $d$ nonzero neurons.*

*Proof.* We can expand the definition of the quadratic network

$$f(x; a, W) = a^\top \sigma(Wx) = \sum_{i=1}^{m} x^\top a_i w_i w_i^\top x.$$

For any $a, W$ such that $f(\cdot; a, W) \equiv f(\cdot)$, we must have $Q = \sum_{i=1}^{m} a_i w_i w_i^\top$. By (a) inequality (2.1) in Thompson (1976) on concave functions of the singular values of sums of matrices (originally proved in Rotfel'd (1969)), we must have

$$\|a\|^2 + \|W\|_F^2 = \sum_{i=1}^{m} a_i^2 + \|w_i\|^2$$
$$\ge 2\sum_{i=1}^{m} \sigma_1(a_i w_i w_i^\top)^{2/3}$$
$$= 2\sum_{i=1}^{m} \sum_{j=1}^{m} \sigma_j(a_i w_i w_i^\top)^{2/3}$$
$$\overset{(a)}{\ge} 2\sum_{j=1}^{d} \sigma_j(Q)^{2/3}.$$

And notice that given an eigendecomposition $(\lambda_1, v_1), \dots, (\lambda_d, v_d)$ of $Q$, this can be achieved by letting $a_i = \mathsf{sgn}(\lambda_i)|\lambda_i|^{1/3}$, and $w_i = |\lambda_i|^{1/3} v_i$ for all $1 \le i \le d$ and $a_i = 0$ and $w_i = 0$ for all $d + 1 \le i \le m$.

$\qquad \square$

# E    DERIVATION AND JUSTIFICATION OF FACT-RFM UPDATE

The simplest fixed-point iteration scheme would be to apply

$$W_{t+1} \leftarrow \sqrt{\mathsf{FACT}_t}, \tag{E.1}$$

aiming for the fixed point

$$W_{t+1}^\top W_{t+1}^\top = \mathsf{FACT}_t.$$

However, this scheme cannot be directly implemented because (E.1) is not necessarily well-defined. In particular, FACT is not necessary p.s.d. when the network is not at a critical point of the loss, so the square root of FACT in (E.1) is not well defined.

In order to fix it, the most natural solution is to symmetrize FACT and instead run the scheme

$$W_{t+1} \leftarrow (\mathsf{FACT}_t\mathsf{FACT}_t^\top)^{1/4},$$

since indeed when $\mathsf{FACT}_t = W_t^\top W_t$ we are at a fixed point with this update.

We experimented with this update, and found good performance with tabular data (this is "no geometric averaging" method reported in Table 1) and parity data, but for the modular arithmetic problem FACT-RFM with this update was unstable and the method often did not converge – especially in data regimes with low signal.

In order to obtain a more stable update, we chose to geometrically average with the previous iterate, as follows:

$$W_{t+1} \leftarrow (\mathsf{FACT}_t(W_t^\top W_t)(W_t^\top W_t)(\mathsf{FACT}_t)^\top)^{1/8},$$

which again has a fixed point when $\mathsf{FACT}_t = W_t^\top W_t$. This yielded improved performance with modular arithmetic while retaining performance with tabular data and parities. Additionally, as we discuss in Section 5, we then discovered that this update has an interpretation as being a close relative of the NFA-RFM update when applied to inner product kernel machines.

# F    PROOFS FOR SECTION 5

We first observe that the updates in FACT-RFM can be written in a convenient form in terms of the dual solution $\alpha$ and the derivatives of the estimator. This lemma does not depend on the kernel being an inner-product kernel.

**Lemma F.1** (Simplified form of FACT for kernel machines). *Let $(X, y)$ be training data fit by a kernel machine with the MSE loss, and let $\alpha$ be first-order optimal coefficients for kernel regression with $\lambda$-ridge regularization. Then the* FACT *can be equivalently computed as*

$$\mathsf{FACT} = \sum_{i,j=1}^n (\frac{\partial}{\partial x} K_W(x, x_j)\mid_{x=x_i})\alpha_j^\top \alpha_i x_i^\top.$$

The proof is by using known first-order optimality conditions for $\alpha$.

Let us prove the convenient expression in Lemma F.1 for the FACT matrix for kernel machines, which can be used to simplify the implementation of FACT-based RFM.

*Proof.* We compute the FACT for the estimator $\hat{f}(x) = \sum_{j=1}^n K_W(x, x_j)\alpha_j$. Substituting the definition of FACT and applying the chain rule, this is

$$\mathsf{FACT} := -\frac{1}{n\lambda}\sum_{i=1}^n (\frac{\partial}{\partial x}\ell(\hat{f}(x), y_i))\mid_{x=x_i} x_i^\top = -\frac{1}{n\lambda}\sum_{i=1}^n (\frac{\partial}{\partial x}\hat{f}(x)\mid_{x=x_i})\ell'(\hat{f}(x_i), y_i)x_i^\top$$

$$= -\frac{1}{n\lambda}\sum_{i,j=1}^n (\frac{\partial}{\partial x}K_W(x, x_j)\mid_{x=x_i})\alpha_j^\top \ell'(\hat{f}(x_i), y_i)x_i^\top,$$

where $\ell' \in \mathbb{R}^c$ denotes the derivative in the first entry. The proof concludes by noting that $\alpha_i = -\frac{1}{n\lambda}\ell'(\hat{f}(x_i), y_i)$ because of the first-order optimality conditions for $\alpha$, proved below in Lemma F.2. $\square$

**Lemma F.2** (Alternative expression for representer coefficients for kernel regression). *Let $(X, Y)$ be training data, and let $\alpha = (K(X, X) + \lambda I)^{-1} Y$ for some $\lambda > 0$. Also let $\ell(\hat{y}, y) = \frac{1}{2}\|\hat{y} - y\|^2$. Then*

$$\alpha_i = -\frac{1}{n\lambda}\ell'(\hat{y}_i, y_i),$$

*where $\hat{y}_i = K(x_i, x)\alpha$, and the derivative $\ell'$ is in the first coordinate.*

*Proof.* Notice that $\ell'(\hat{y}_i, y_i) = \hat{y}_i - y_i$. So

$$\begin{aligned}
\ell'(\hat{y}_i, y_i) &= [K\alpha]_{i,*} - y_i \\
&= K(K + n\lambda I)^{-1} y_i - y_i \\
&= -n\lambda(K + \lambda I)^{-1} y_i \\
&= -n\lambda\alpha_i \,.
\end{aligned}$$

$\square$

*Remark* F.3. A statement of this form relating the representer coefficients to the loss derivatives at optimality is more generally true beyond the MSE loss, but we do not need it here.

Finally, we can prove Proposition 5.1.

**Proposition F.4** (Restatement of Proposition 5.1). *Suppose the kernel is an inner-product kernel of the form $K_W(x, x') = k(x^\top M x')$, where $M = W^\top W$. Then, we may write the* AGOP *and the* FACT *matrices explicitly as:*

$$\mathsf{AGOP} = \sum_{i,j=1}^{n} \tau(x_i, x_j, M) M x_i \alpha_i^\top \alpha_j x_j^\top M^\top \,,$$

$$\mathsf{FACT} \cdot M^\top = \sum_{i,j=1}^{n} k'(x_i^\top M x_j) M x_i \alpha_i^\top \alpha_j x_j^\top M^\top \,,$$

*where $\tau(x_i, M, x_j) := \frac{1}{n}\sum_{l=1}^{n} k'(x_l^\top M x_i) k'(x_l^\top M x_j)$.*

*Proof.* The expressions can be derived by plugging in the expansion $\hat{f}(x) = \sum_{j=1}^{n} K(x, x_i)\alpha_i$.

For AGOP, we start from its expression in Ansatz (NFA), and obtain

$$\begin{aligned}
\mathsf{AGOP} &= \sum_{i=1}^{n}(\nabla_x \sum_{j=1}^{n} K_W(x, x_j)\alpha_j)(\nabla_x \sum_{l=1}^{n} K_W(x, x_l)\alpha_l)^\top \\
&= \sum_{i,j,l=1}^{n} k'(x_j^\top M x_i)k'(x_l^\top x_i)(M x_j \alpha_j^\top)(M x_l \alpha_l^\top)^\top \\
&= \sum_{i,j=1}^{n} \tau(x_i, M, x_j) M x_i \alpha_i^\top \alpha_j x_j^\top M^\top \,.
\end{aligned}$$

For FACT, we start from the expression in Lemma F.1:

$$\begin{aligned}
\mathsf{FACT} \cdot M^\top &= \sum_{i,j=1}^{n}(\frac{\partial}{\partial x} K_W(x, x_j)\mid_{x=x_i})\alpha_j^\top \alpha_i x_i^\top M^\top \\
&= \sum_{i,j=1}^{n} k'(x_i^\top M x_j) M x_j \alpha_j^\top \alpha_i x_i^\top M^\top \,.
\end{aligned}$$

$\square$

## G    EXPERIMENTAL RESOURCE REQUIREMENTS

The following timings are for one A40 48GB GPU. The tabular data benchmark experiments in Table 1 take under 1 GPU-hour to run. The synthetic benchmark task of Figure 7 on which FACT and NFA are uncorrelated takes under 1 GPU-hour to run. The arithmetic experiments in Figure 5 and 6 take under 1 GPU-hour to run. The ReLU MLP experiments on MNIST and CIFAR-10 in Figures 2, 8, 9, and 10 take under 50 GPU-hours to run. The sparse parity experiments in Figure 4 and Figure 3 take under 1 GPU-hour to run. The deep linear network experiments in Figure 11 take under 2 GPU-hours to run. Additionally, debugging code and tuning hyperparameters took under 200 GPU-hours to run.

## H    LLM USAGE

LLMs were used only as AI coding assistants and to help polish some of the writing in the paper, and were not used for research ideation.

