# OpenReview forum: "FACT: a first-principles alternative to the Neural Feature Ansatz for how networks learn representations"
_ICLR.cc/2026/Conference — ICLR 2026 Poster_

### Official Review · Reviewer_f429 · 2025-10-21

**Soundness:** 3
**Presentation:** 3
**Contribution:** 3
**Rating:** 8
**Confidence:** 4

**Summary:**

The authors present FACT, a self-consistent condition for L2-regularized neural networks. This condition provides a more theoretically grounded alternative to the NFA conjecture, establishing a connection to (locally) minimum-norm solutions. The authors demonstrate both empirically and theoretically that FACT aligns more closely with the behavior of trained fully connected networks than existing NFA-based approaches.

On the empirical side, the authors compare FACT-RFM with NFA-RFM, showing that the two exhibit similar phenomena, such as phase transitions and grokking, across various tasks. Theoretically, they prove a case, under a minimum-norm objective, where the NFA conjecture fails while FACT continues to hold.

**Strengths:**

The main contribution of this work is to extend the empirically motivated NFA conjecture to a more theoretically grounded framework based on (local) minimum-norm conditions. This perspective helps bridge the gap between the NFA line of research and the broader literature on L2-regularized solutions, highlighting RFM methods as an interesting alternative to gradient-based training.

The paper is clearly written, and the experiments are illustrative. To the best of my knowledge, the proofs are correct. Overall, this is a solid submission with only minor weaknesses.

**Weaknesses:**

The paper could benefit from a stronger early narrative explaining why FACT is a meaningful measure — specifically, how its foundation in (local) minimum-norm behavior motivates the framework. Although Sections 5 and 6 provide this perspective more clearly, the earlier parts of the paper introduce it less directly. For instance, establishing the connection earlier would help clarify how the low-rank bias and neural collapse phenomena [1] in Figure 2 relate to L2 regularization [2-3], and likewise how the grokking behavior in Figure 5 connects to L2-driven dynamics [4].

As a minor comment, the acronym “FACT” has appeared previously in the literature [5]. In addition, the name emphasizes self-consistency at convergence but does not reflect the central role of L2 regularization, which is arguably more important.

 1. X. Y. Han et al., 2022 Neural Collapse Under MSE Loss: Proximity to and Dynamics on the Central Path
 2. T. Galanti et al., 2024 SGD and Weight Decay Secretly Minimize the Rank of Your Neural Network
 3. E. Zangrando et al.,2024 Neural Rank Collapse: Weight Decay and Small Within-Class Variability Yield Low-Rank Bias
 4. Lyu et al., 2024 Dichotomy of Early and Late Phase Implicit Biases Can Provably Induce Grokking
 5. Yubin Qin et al., 2023 FACT: FFN-Attention Co-optimized Transformer Architecture with Eager Correlation Prediction

**Questions:**

How different is the minimum-norm solution from the solution obtained by FACT-RFM in practice? Are they generally equivalent, or are there theoretical or empirical cases the authors are aware of where they diverge?

---

> ### Author Response · Authors · 2025-11-28
>
> ## Note: we responded after the OpenReview leak, so unfortunately there was no chance for discussion
>
> --
>
> Thank you for your detailed comments. Also thank you for your positive appraisal of our manuscript. We respond to your questions below.
>
> > The paper could benefit from a stronger early narrative explaining why FACT is a meaningful measure — specifically, how its foundation in (local) minimum-norm behavior motivates the framework....
>
> We added more note of this to the introduction and tried to emphasize more that we are uniting minimum-norm with NFA.
>
> > As a minor comment, the acronym “FACT” has appeared previously in the literature [5]....
>
> Thank you for making us aware of this.
>
> > How different is the minimum-norm solution from the solution obtained by FACT-RFM in practice? Are they generally equivalent, or are there theoretical or empirical cases the authors are aware of where they diverge?
>
> This is a good question. We did not check whether they diverge. However, notice that FACT-RFM is effectively a locally-minimum-norm condition. This means that at the very least FACT-RFM will converge to a locally minimum-norm solution.

---

### Official Review · Reviewer_Z58e · 2025-10-30

**Soundness:** 4
**Presentation:** 4
**Contribution:** 2
**Rating:** 4
**Confidence:** 4

**Summary:**

The authors propose a principled derivation of first order optimality conditions of neural network matrices at convergence. They show that this explains many of the same phenomena as neural feature ansatz.

**Strengths:**

The paper presents a rigorous and principled theoretical derivation for previously empirically observed (or conjectured) phenomena around neural feature ansatz.

They prove their key theorem which gives a nice intuition for why this occurs.

They provide solid experiments that support the theory.

They have an insightful limitations figure that shows how their theory only holds at convergence (as opposed to NFA)

**Weaknesses:**

There is a whole field of literature on identifiability and representation learning (see here for a start: https://link.springer.com/article/10.1007/s10463-023-00884-4) that studies the representations learned by neural networks at convergence. I expect the authors to add those references and discuss the relation of their work to that field.

Fig2 caption: define what it means to train until 'interpolation'

lines 198-200: unclear argument, the following sentences are just algebraic manipulations

Many of the experimental setups read like: 'we found it works for these hyperparameters' – it would be great to see some ablation studies, e.g., how do think still/no longer work with different learning rate, dimensionality, weight decay etc.

236: 'akin to feature leaning in neural networks' – I think this is a key overstatement (also in the NFA literature), all that is happening here is learned linear filtering on the inputs. This is very different from *nonlinear* feature learning in neural networks. Please tone this down.

262 please write out FACT again below this equation, I think that would show how this is almost tautologically true, that FACT-RFM performs better because it just looks like a gradient step of W w.r.t. to the loss

**Questions:**

Please take all questions below as call for action to make your paper more clear:

Why is eNFA called equivariant? Please explain in paper.

On the first page, NFA is the gradient with respect to inputs, on page 3 it is w.r.t. to hidden activations. Which of the two is it?

206: how sensitive is this to different values of weight decay?

207: what correlation are you measuring? since NFA and AGOP are only proportional I would expect Spearman. However, for FACT, it is an equality so I would expect Pearson or even variance explained; Also, why look at sqrt? This can be avoided with rank correlation.

fig5: how exactly does NFA or FACT explain grokking? or is this just an observation about behavior of the system? fig 8 shows that FACT only holds at convergence… what do we learn here?

---

> ### Author Response · Authors · 2025-11-28
>
> ## Note: we responded after the OpenReview leak, so unfortunately there was no chance for discussion
>
> --
>
>
> Thank you for your detailed comments. Also thank you for your positive appraisal of our theorems, experiments, and limitations. We seek to address your concerns below.
>
> > There is a whole field of literature on identifiability and representation learning (see here for a start... I expect the authors to add those references and discuss the relation of their work to that field.
>
> Thank you for the references. We have incorporated them, as well as a discussion of the relation of our work to that field in Section 1.1.
>
>
> > Fig2 caption: define what it means to train until 'interpolation'
>
> We operationalize this as batch train loss $\leq 10^{-3}$ (see Appendix A). We have edited the caption of the figure to clarify this.
>
>
> > lines 198-200: unclear argument, the following sentences are just algebraic manipulations
>
> We modified the text for more clarity.
>
> > Many of the experimental setups read like: 'we found it works for these hyperparameters' – it would be great to see some ablation studies, e.g., how do think still/no longer work with different learning rate, dimensionality, weight decay etc.
>
> We added some ablations with different number of training samples and weight decays in the appendix.
>
> > 236: 'akin to feature leaning in neural networks' – I think this is a key overstatement (also in the NFA literature), all that is happening here is learned linear filtering on the inputs. This is very different from nonlinear feature learning in neural networks. Please tone this down.
>
> We toned it down by specifying it is about feature learning within a layer of neural networks.
>
> > 262 please write out FACT again below this equation, I think that would show how this is almost tautologically true, that FACT-RFM performs better because it just looks like a gradient step of W w.r.t. to the loss
>
> It is not look a gradient step of W w.r.t the loss. Instead that becomes
> $W^{\top} W \gets (W^{\top} \nabla_W \mathcal{L}(W) (\nabla_W \mathcal{L}(W))^{\top} W)^{1/2}$
>
> > Why is eNFA called equivariant? Please explain in paper.
>
> This is the name given by Ziyin et al, since they modify NFA so that it works even when the network and the loss functions are modified by linear transformation. We have clarified this in the paper.
>
> > On the first page, NFA is the gradient with respect to inputs, on page 3 it is w.r.t. to hidden activations. Which of the two is it?
>
> It is gradient with respect to input to the layer with matrix W. If W is in the middle of the network, then the inputs are the hidden activations. We have clarified this in the paper.
>
> > 206: how sensitive is this to different values of weight decay?
>
> We tried varying the weight decay (10^{-3} and 10^{-4}) and added results in the appendix. The result does not seem sensitive to these values, which are chosen in typical hyperparameter ranges.
>
> > 207: what correlation are you measuring?...
>
> We are measuring Pearson correlation, which normalizes so proportionality factors don’t affect it. The square root (power $s = 1 / 2$) is suggested by prior work on NFA as what works best in practice – in Appendix C we study the best power for linear networks of different depths. We have clarified this.
>
> > fig5: how exactly does NFA or FACT explain grokking? or is this just an observation about behavior of the system? fig 8 shows that FACT only holds at convergence… what do we learn here?
>
> This is a surprising observation about the behavior of the system. Despite FACT being only proved at convergence, it can be used to power the Recursive Feature Machine algorithm and obtain the same grokking behavior on arithmetic tasks as NFA. Thus, we argue that many of the predictions about feature learning that were done with NFA can instead be done with the more principled FACT. This question is discussed in our discussion section.

---

### Official Review · Reviewer_kUP6 · 2025-10-31

**Soundness:** 4
**Presentation:** 4
**Contribution:** 4
**Rating:** 8
**Confidence:** 4

**Summary:**

This paper derives an alternative to the Neural Feature Ansatz by investigating first-order optimality conditions of a loss function with L2 weight penalty. The alternative, FACT, performs similarly to the NFA in applications, but correlates better to solutions found in neural networks. The paper further shows a synthetic dataset for which NFA predictions fail completely while FACT predictions remain accurate. Overall these results show that FACT is a useful and theoretically principled alternative to NFA.

**Strengths:**

Derives an expression, FACT from first order conditions which is reminiscent of the NFA, thereby providing a theoretical basis for the method (the justification of NFA by contrast being empirical). This is a great contribution.

The experiments show that FACT behaves similarly to NFA in applications to tabular data, sparse parity, and grokking. These results show that FACT is approximately as useful as NFA.

The result showing that NFA predictions can break down while FACT predictions remain is particularly nice in highlighting the contribution of the work.

Overall the paper is clear and easy to read, with well made figures.

**Weaknesses:**

While the proposed FACT method may have a theoretical justification, it does not clearly improve on the NFA method in the applications considered here. The paper could be strengthened by including a case where FACT outperforms NFA on more than a synthetic task. But I do not view this as an essential addition to the paper.

**Questions:**

Are there any intuitions for why NFA correlates well with weights throughout learning but FACT does not?

---

> ### Author Response · Authors · 2025-11-28
>
> ## Note: we responded after the OpenReview leak, so unfortunately there was no chance for discussion
>
> --
>
> Thank you for your detailed review. We are very glad that you appreciated our results and think that they are a valuable contribution. Please see below for our answers to your questions.
>
> > While the proposed FACT method may have a theoretical justification, it does not clearly improve on the NFA method in the applications considered here....
>
> This is a good point, and it would indeed be great to have a real-life task where there is such a separation. This seems like it would fit well with a follow-up paper that studies why sqrt(FACT FACT^T) has better correlation throughout training with the ground truth than FACT, and tries to find even better alternatives.
>
> > Are there any intuitions for why NFA correlates well with weights throughout learning but FACT does not?
>
> This is a great question, and we do not have a well-formed intuition right now. We know from Section 5 that we should expect sqrt(NFA) to be roughly equivalent to sqrt(FACT * W^TW) in the case of training inner product kernel machines. Therefore NFA has an extra "W^TW" term implicitly in its definition, so information about W^TW is "leaked" in the definition of NFA. This may be the reason for a larger correlation throughout, but we are not sure.

---

### Official Review · Reviewer_NSpx · 2025-10-31

**Soundness:** 3
**Presentation:** 3
**Contribution:** 2
**Rating:** 4
**Confidence:** 3

**Summary:**

The paper introduces the Features at Convergence Theorem (FACT), a theoretical alternative to the Neural Feature Ansatz, the idea that a layer’s feature covariance correlates with the average gradient outer product. FACT is derived from first-order optimality conditions that must hold at convergence, and thus is more theoretically grounded. Empirical tests show that FACT reproduces key learning behaviours — including grokking, phase transitions, and sparse parity learning — similar to NFA. FACT matches NFA’s predictions in most settings but remains valid where NFA fails.

**Strengths:**

- Clarity and simplicity. The core identity (FACT) follows directly from first‑order optimality with L2 weight decay, relating a layer’s feature Gram $W^\top W$ to an empirical average of loss‑gradients times activations. It’s architecture‑agnostic under the mild requirement that the model depends on a weight matrix multiplication (Equation 2.1 and Theorem 3.1). This gives a simple target that any trained model with weight decay must satisfy.
- Forward and backward views. Besides the forward statement for $W^T W$, they derive a backward counterpart for $WW^T$ (bFACT), giving traction on both right and left singular spaces of a layer (Remark 3.4).
- Empirical alignment at convergence across realistic vision setups. In 5‑layer ReLU MLPs on MNIST/CIFAR‑10, Pearson correlation between the two sides of FACT and the learned $W^T W$ rises to $\sim1$ at convergence (Figure 2), often exceeding NFA and eNFA. This makes FACT a strong descriptor of terminal‑phase features in practice.

**Weaknesses:**

- FACT's correlation with $W^T W$ is typically low during most of training and spikes only near interpolation (Figure 8). This makes FACT relatively uninformative about how features emerge, arguably the purpose of studying these quantities.
- Mathematical novelty is modest. The main theorem is essentially the stationarity condition written in feature‑centric form.
Applies to the terminal phase only.
Kernel analysis is clarifying but incremental. The inner‑product kernel derivations neatly align the two updates $\tau$ vs $k'$, but the algebra is standard once you adopt the kernelised view. It explains why NFA often works without yielding new predictive dynamics (Section 5).
- Breadth of architectures/losses is limited. Core validations focus on MLPs with MSE; connections to cross‑entropy (with weight decay), convs/transformers, and decoupled weight decay (AdamW) are not systematically explored. eNFA is included but comparisons remain relatively narrow (Sections 3-4).

**Questions:**

1. FACT’s correlation with (W^\top W) is low through most of training and spikes only near interpolation (Figure 8). What intermediate quantity (e.g. a the $\sqrt{\text{FACT}\cdot\text{FACT}^T} $ you note) best tracks feature emergence earlier, and can you motivate it theoretically rather than empirically?
2.    Since FACT follows from first-order optimality, can you predict when alignment will occur, not just that it must at convergence? Any falsifiable prediction for the timing of grokking/phase transitions beyond endpoint agreement?
3. Given that Theorem 3.1 is a stationarity rewrite, what’s the new predictive leverage beyond endpoint consistency, for example,  conditions under which FACT will disagree with NFA before convergence in natural (non-adversarial) settings?

---

> ### Author Response · Authors · 2025-11-28
>
> ## Note: we responded after the OpenReview leak, so unfortunately there was no chance for discussion
>
> --
>
> Thank you for your detailed review, with many interesting questions. And thank you for your positive comments on the simplicity, generality, and realism of our results. Below, we address your concerns.
>
>
> > FACT's correlation with W T W is typically low... purpose of studying these quantities.
>
> This is a good point and we tried to emphasize this in our discussion. Nevertheless, we disagree that this means that FACT is uninformative. Instead, we make the surprising observation that training recursive feature machines with FACT recovers feature learning phenomena, so FACT does capture important aspects of how features emerge.
>
> > Mathematical novelty is modest....
>
> Our goal with this paper is to better understand neural networks, and we believe that the easier to understand the math is, the better.
>
> The main insight of this paper is that there is a connection between the Neural Feature Ansatz and stationarity conditions. This insight is new to the literature, and leads to a better understanding of feature learning in neural networks. Once we have this insight, the math to communicate it follows elegantly.
>
>
>
> We also proved results in this paper that are mathematically involved and require developing new techniques. The proof that NFA can be uncorrelated with the ground truth is elaborate – see Appendix D. It requires characterizing minimum-norm solutions to quadratic networks, which is of independent interest, and our proof there uses techniques from matrix analysis.
>
>
> > It explains why NFA often works without yielding new predictive dynamics (Section 5).
>
> Our work yields new predictive dynamics. Even though the NFA often agrees with FACT, their relationship is imperfect (see Figure 6 and Section 6). This means that FACT-RFM is a different algorithm from NFA-RFM.
>
> > Breadth of architectures/losses is limited....
>
> This paper primarily has a theoretical contribution showing a connection between NFA and stationarity conditions. In the revision, we added some ablation experiments in the appendix showing that the results are consistent when varying the number of training samples and weight decay. We also recall that we have several experiments in this paper beyond these: (a) the recursive feature machine experiments on parities, tabular data, and modular arithmetic, (b) the comparison between FACT and NFA in the case of inner product kernels, (c) the strong separation between FACT and NFA, and (d) experiments on deep linear networks.
>
>
> > FACT’s correlation with (W^\top W) is low through most of training and spikes only near interpolation...
>
> This is a good question. As we noted we find experimentally that the sqrt(FACT FACT^T) quantity tracks feature emergence earlier. We are not sure why this is the case, and do not yet have a theoretical motivation. This seems like a good open problem for future work.
>
>
> > Since FACT follows from first-order optimality, can you predict when alignment will occur...
>
> Our results only predict that alignment will occur at convergence. However, using FACT with recursive feature machines recovers feature learning behaviors such as grokking and phase transitions beyond endpoint agreement. It would be interesting to analyze the dynamics in cases such as modular arithmetic, but we think it is outside of the scope of the current paper.
>
>
> > Given that Theorem 3.1 is a stationarity rewrite...
>
> The connection between NFA and stationarity conditions (a) allows us to predict that NFA & FACT will agree when the similarity measures \tau and k’ in Section 5 are proportional,and  (b) allows us to show that feature learning behaviors with recursive feature machines can be observed using the theoretically principled FACT instead of NFA. This paper argues that FACT should be adopted as a principled alternative to NFA.
>
> We hope that we addressed your concerns with our responses.

---

### Meta-Review · Area_Chair_fQrU · 2026-01-08

**Summary:**

###### Reviewer NSpx

(1) The theory applies only at convergence; (2) the mathematical development is simple; (3) the scope of experiments is limited: MLPs with MSE.

###### Reviewer kUP6

(1) A more significant example of real-life separation between NFA and FACT would be helpful.

###### Reviewer Z58e
(1) There is a lack of references to the literature on identifiability in representation learning; (2) there is a lack of ablation studies or justification for hyperparameter choices; (3) the scope of feature learning in this work is limited to learned linear filtering (cf. nonlinear feature learning) because of the first-order construction, despite the claim to cover feature learning in general.

###### Reviewer f429

(1) A generic concern about lack of clarity.

**Reviewer Concerns:**

##### Addressed

###### Reviewer NSpx (2)

New mathematical techniques are not a claimed contribution of the work. Instead, what is claimed as a new view on NFA through first order optimality conditions, which is provided via FACT.

###### Reviewer NSpx (3)

This would be satisfactorily addressed in a revision if the authors more clearly outlined the scope of their theory (where it does not apply). The authors already clearly mark in Figure 1 the scope of architectures covered.

###### Reviewer kUP6 (1)

The reviewer conveyed that this concern was not a dealbreaker. I concur.

###### Reviewer f429 (1)

Since the reviewer's claim was non-specific, there is nothing to address.

##### Outstanding

###### Reviewer NSpx (1)

The authors responded in the rebuttal that "FACT does capture important aspects of how features emerge" and "[the] work yields new predictive dynamics" but the theory is about optimality, not dynamics. Indeed, I found the paper was confused about this contribution, claiming in the introduction that "the NFA conjecture lacks first-principles backing for why it should necessarily hold during training," yet the FACT is a property that does not speak to training either, as it is an optimality condition. Describing the *emergence* of feature learning requires a theoretical statement about a dynamical phenomenon, like time to learn features, or the order in which features are learned. These are not provided here.

The authors additionally claimed that the kernel machine view captures dynamical phenomenon, but no guarantees about these phenomenon are provided by the theory. Instead, the theory provides a heuristic construction for a kernel machine. This is a different, empirical contribution from the theory.

###### Reviewer Z58e (1)

The authors included the reference but contested the relationship between feature learning and identifiability. I think this was insufficient to deflect discussion.

###### Reviewer Z58e (2)

The authors responded to this with a sweep over two hyperparameters in the appendix. This is limited with respect to the scope of investigation that the reviewer asked for.

###### Reviewer Z58e (3)

The authors did not address the scope of feature learning being limited in their work to *linear* feature learning. There is extensive work on nonlinear feature learning in neural networks that is not cited ([Chizat, Oyallon, & Bach (2019)](https://arxiv.org/abs/1812.07956); [Yang & Hu (2021)](https://arxiv.org/abs/2011.14522); [Mei, Montanari, & Nguyen (2018)](https://arxiv.org/abs/1804.06561)). Thus, the reviewer's comment is not addressed.

**Reviewer Scores:**

I would expect Reviewer NSpx to have modestly updated their score after further discussion about valid extensions of the theory (other losses) and some expectations about applications  (or failures thereof) beyond the scope of the theory (other architectures like transformers, with data dependent weights).

I believe Reviewer Z58e would have not updated their score because their concerns were insufficiently addressed, though their concerns are about messaging rather than the details of the contribution.

I would expect the remaining reviewers' scores to stay put at 6. Because of this, I weakly recommend acceptance.

---

### Decision · Program_Chairs · 2026-01-26

Accept (Poster)